# Cardiomyocyte-Targeting Peptide to Deliver Amiodarone

**DOI:** 10.3390/pharmaceutics15082107

**Published:** 2023-08-09

**Authors:** Maliha Zahid, Beth Weber, Ray Yurko, Kazi Islam, Vaishavi Agrawal, Jack Lopuszynski, Hisato Yagi, Guy Salama

**Affiliations:** 1Department of Cardiovascular Diseases, Mayo Clinic, Rochester, MN 55905, USA; lopuzinski.jack@mayo.edu; 2Pittsburgh Heart, Lung, Blood, and Vascular Medicine Institute, Division of Cardiology, Department of Medicine, University of Pittsburgh School of Medicine, University of Pittsburgh Medical Center, Pittsburgh, PA 15213, USA; bethgabris@gmail.com (B.W.); gsalama@pitt.edu (G.S.); 3Peptide Synthesis Facility, University of Pittsburgh, Pittsburgh, PA 15219, USA; yurko@pitt.edu (R.Y.); kazi@pitt.edu (K.I.); 4Dietrich School of Arts and Sciences, University of Pittsburgh, Pittsburgh, PA 15260, USA; vaa28@pitt.edu; 5Department of Developmental Biology, University of Pittsburgh, Pittsburgh, PA 15201, USA; yagi@pitt.edu

**Keywords:** cardiac targeting peptide, cell penetrating peptides, amiodarone, atrial fibrillation

## Abstract

Background: Amiodarone is underutilized due to significant off-target toxicities. We hypothesized that targeted delivery to the heart would lead to the lowering of the dose by utilizing a cardiomyocyte-targeting peptide (CTP), a cell-penetrating peptide identified by our prior phage display work. Methods: CTP was synthesized thiolated at the N-terminus, conjugated to amiodarone via Schiff base chemistry, HPLC purified, and confirmed with MALDI/TOF. The stability of the conjugate was assessed using serial HPLCs. Guinea pigs (GP) were injected intraperitoneally daily with vehicle (7 days), amiodarone (7 days; 80 mg/kg), CTP–amiodarone (5 days; 26.3 mg/kg), or CTP (5 days; 17.8 mg/kg), after which the GPs were euthanized, and the hearts were excised and perfused on a Langendorff apparatus with Tyrode’s solution and blebbistatin (5 µM) to minimize the contractions. Voltage (RH237) and Ca^2+^-indicator dye (Rhod-2/AM) were injected, and fluorescence from the epicardium split and was captured by two cameras at 570–595 nm for the cytosolic Ca^2+^ and 610–750 nm wavelengths for the voltage. Subsequently, the hearts were paced at 250 ms with programmed stimulation to measure the changes in the conduction velocities (CV), action potential duration (APD), and Ca^2+^ transient durations at 90% recovery (CaTD_90_). mRNA was extracted from all hearts, and RNA sequencing was performed with results compared to the control hearts. Results: The CTP–amiodarone remained stable for up to 21 days at 37 °C. At ~1/15th of the dose of amiodarone, the CTP–amiodarone decreased the CV in hearts significantly compared to the control GPs (0.92 ± 0.05 vs. 1.00 ± 0.03 ms, *p* = 0.0007), equivalent to amiodarone alone (0.87 ± 0.08 ms, *p* = 0.0003). Amiodarone increased the APD (192 ± 5 ms vs. 175 ± 8 ms for vehicle, *p* = 0.0025), while CTP–amiodarone decreased it significantly (157 ± 16 ms, *p* = 0.0136), similar to CTP alone (155 ± 13 ms, *p* = 0.0039). Both amiodarone and CTP–amiodarone significantly decreased the calcium transients compared to the controls. CTP–amiodarone and CTP decreased the CaTD_90_ to an extent greater than amiodarone alone (*p* < 0.001). RNA-seq showed that CTP alone increased the expression of DHPR and SERCA2a, while it decreased the expression of the proinflammatory genes, NF-kappa B, TNF-α, IL-1β, and IL-6. Conclusions: Our data suggest that CTP can deliver amiodarone to cardiomyocytes at ~1/15th the total molar dose of the amiodarone needed to produce a comparable slowing of CVs. The ability of CTP to decrease the AP durations and CaTD_90_ may be related to its increase in the expression of Ca-handling genes, which merits further study.

## 1. Introduction

Atrial fibrillation (afib) is the most common arrhythmia in adults with an estimated prevalence of 46.3 million people worldwide [1]. It affects 1–2% of adults in the USA, with an estimated 5.7 million patients with afib [2]. Data from the Framingham heart study have documented an over threefold increase in afib prevalence over the last 50 years [3]. These already high rates are estimated to double by year 2050 due to an aging population and the increased prevalence of risk factors like coronary artery disease, hypertension, obesity, and congestive heart failure [3]. The superiority of rhythm over rate control in afib has been demonstrated in multiple randomized control trials [4,5,6]. Although amiodarone is the most efficacious anti-arrhythmic drug with demonstrated superiority over multiple other anti-arrhythmics like propafenone [7,8], sotalol [7,9,10], and dronedarone [11,12], it is grossly underutilized and remains a second-line therapy for maintenance of normal sinus rhythm in patients with afib. This is due to amiodarone being a highly lipophilic drug with a large volume of distribution [13], a long half-life, uptake by noncardiac tissues like lungs, liver, thyroid, skin and ocular tissue, leading to multiple off-target and potentially life-threatening toxicities, thus limiting its use to patients over 75 years of age or patients with limited life expectancies [14]. Additionally, amiodarone is the drug of choice for ventricular tachycardia [15,16], as well as for reducing defibrillator shocks in patients with cardiomyopathies who have an implanted automatic intravascular defibrillation [16,17]. Hence, amiodarone is a classic demonstration of the Achilles heel of both cardiac diagnostics and therapeutics suffering from a lack of cardiac specific vector(s) and lack of targeted delivery.

The cell plasma membrane is a semipermeable barrier that is essential for cell integrity and survival but, at the same time, presents a barrier to delivery of cargo. Hence, the ability of certain proteins to cross the cell membrane barriers was met with great enthusiasm. In 1988, two separate groups demonstrated the ability of the trans-activator of transcription (Tat) protein of the human immunodeficiency virus (HIV) to enter cultured cells and promote viral gene expression [18,19]. Mapping of the domain responsible for this transduction ability led to the identification of the first cell-penetrating peptide, Tat, corresponding to the 11 amino acid basic domain of HIV-1 Tat protein. Subsequently, it was demonstrated that Tat fused to β-galactosidase and injected intraperitoneally into mice was internalized into multiple cell types including liver, heart, lung, and kidney, and even crossed the blood–brain barrier delivering β-galactosidase in a functional form, highlighting the potential of these peptides as vectors [20].

The ability of cationic or hydrophobic cell-penetrating peptides to transduce a wide variety of tissue types in vivo limits their clinical utility because of their lack of cell specificity, leading to a higher chance of off-target adverse side effects. Phage display using libraries of various lengths and various bacteriophage strains have been successfully utilized to identify tissue-specific penetrating peptides [21,22,23,24,25]. In our prior work, we employed a combinatorial in vitro and in vivo M13 phage display methodology [21] to identify a 12-amino-acid, synthetic, mildly basic, and non-naturally occurring peptide (NH_2_-APWHLSSQYSRT-COOH) that we termed the Cardiac-Targeting Peptide (CTP), due to its ability to specifically target cardiomyocytes in vivo after tail-vein injection in mice [21,26]. Our detailed biodistribution studies showed that peak uptake occurred as early as 15 min with the complete disappearance of the fluorescently labeled CTP by 6 h [27]. CTP does not cross the blood–brain barrier and is not taken up by striated muscle, spleen, lungs, etc. CTP appears to be excreted predominantly by a renal followed by a hepatobiliary mechanism [27]. Our findings were replicated by at least three independent laboratories with published work showing that indeed CTP is cardiomyocyte-specific [28,29,30]. It has been used to deliver photosensitizers to the cardiomyocytes for the targeted ablation of afib in a sheep model, sparing damage to “innocent bystander” cardiac myofibroblasts and endothelial cells [28]. Additionally, exosomes labeled with CTP and loaded with anti-RAGE (Receptor for Advanced Glycation End-products) siRNA were able to ameliorate myocarditis in a rat model [30]. CTP is not limited to mice, as its efficacy as a vector has been demonstrated in sheep [28] and rat models [30] to deliver myriad cargoes. Indeed, our own lab showed that human heart tissue explanted from patients undergoing heart transplants could be successfully transduced with fluorescently labeled CTP with uptake of the peptide by normal cardiomyocytes while sparing the fibroblasts that make up the intervening scar tissue in these diseased hearts. Moreover, the uptake in cardiomyocytes was not due to an increase in plasma membrane permeability as demonstrated by a lack of uptake of Evans blue [31].

In the current manuscript, we present studies showing the successful conjugation of amiodarone to the N-terminus of CTP via a disulfide linker. We tested our conjugate in guinea pigs (GPs) to compare the changes in the calcium-handling equivalent to that seen with amiodarone but at a markedly decreased (1/15th) total dose of amiodarone. A surprise finding was the additional effects on calcium transients that were attributed to the CTP portion of the CTP–amiodarone conjugate and not amiodarone. This led us to perform gene expression studies that showed that CTP had additional salutary anti-inflammatory effects and changes in calcium-handling genes. Our findings lead us to conclude that CTP is not simply an inert vector but has additional, potentially beneficial, biological effects on cardiomyocytes.

## 2. Methods

### 2.1. CTP Synthesis and Conjugation to Amiodarone

Solid phase peptide synthesis (SPPS) of CTP was accomplished on a Liberty CEM microwave synthesizer using Fmoc/tBu chemistry and Oxyma pure coupling protocols on Rink amide resin. Upon completion of the CTP-peptide chain assembly, the free N-terminal amino group was manually conjugated to 3-(Tritylthio)propionic acid with TBTU/HOBt/DIPEA in DMF. The thiol-modified CTP peptide resin was then cleaved with trifluoroacetic acid (TFA) + scavengers followed by isolation of the crude product by precipitation in diethyl ether (Et_2_O). The crude CTP–thiol was then purified by preparative C-18 RP-HPLC on a Waters Delta Prep 4000 (Framingham, MA, USA) chromatography system followed by lyophilization. The purified CTP–thiol was then reacted with 2,2’-Dithiobis(5-nitropyridine) (DTNP) in 80% TFA(aq) for 15 min and then dried to a film using a stream of nitrogen. Dissolution of the 5-Nitro-2-Pyridinesulfenyl (pNpys) activated CTP analogue in 50% Trifluoroethanol/0.1% TFA was followed by preparative C-18 RP-HPLC purification on a Waters Delta Prep 4000 chromatography system and lyophilization to a dry powder.

Amiodarone hydrochloride was chloroalkylated at the tertiary nitrogen using 3-Chloro-1-propanethiol forming a thiolated quaternary ammonium salt. Amiodarone–thiol was then purified by preparative C-18 RP-HPLC purification on a Waters Delta Prep 4000 chromatography system and then lyophilized. The purified amiodarone–thiol was then reacted with pNpys-CTP in ammonium acetate buffered solution at pH 4 for 2 h. Preparative C-18 RP-HPLC purification on a Waters Delta Prep 4000 chromatography system was followed by lyophilization to a dry powder [32]. Analytical C-18 RP-HPLC characterization of Amiodarone-SS-CTP on a Waters Alliance chromatography system was performed followed by mass analysis on a Bruker UltraFlextreme MALDI TOF/TOF mass spectrometer (Framingham, MA, USA) to confirm the expected mass and purity of the final product. Please see the publication by R. Yurko and colleagues for the step-by-step detailed synthesis protocol [32].

### 2.2. CTP Targeting of Cardiomyocytes

A wildtype Sprague Dawley rat heart was mounted on a Langendorff perfusion system and perfused with 50 μL of a 1 mM solution of Cy5.5 labeled CTP. After the bolus injection, the heart was fixed in paraformaldehyde, embedded, cryosectioned with 7 μm thick sections obtained that were cross-stained with WGA-488 (wheat germ agglutinin labeled with Alexa fluoro dye 488) to stain for myofibroblasts and DAPI, a nuclear stain, and confocal imaging was performed.

### 2.3. Conjugate Stability Studies

Freshly synthesized/purified CTP-amiodarone conjugate was HPLC characterized after lyophilization as the baseline measure. A small amount (1 mg/mL) was dissolved in the buffer used for injecting animals and placed in a water bath at 37 °C for up to 22 days. A small aliquot was taken at regular intervals, and an HPLC was run on the Waters Delta Prep 4000 chromatography system.

### 2.4. Guinea Pig Studies

All animal protocols were approved by the University of Pittsburgh’s institutional animal care and use committee prior to undertaking any experimentation. Adult male 6-week-old GPs (250–350 g) were injected intraperitoneally daily with vehicle (7 days), amiodarone (7 days; 80 mg/kg), CTP–Amiodarone at 1/10th the molar dose of amiodarone (5 days; 26.3 mg/kg), or CTP at 1/10th the molar dose of amiodarone (5 days; 17.8 mg/kg). At the end of the study, the GPs were euthanized, the hearts excised and perfused on a Langendorff apparatus with Tyrode’s solution containing (in mM): NaCl (130), KCl (4), MgSO_4_ (1.2), NaHCO_3_ (25), Glucose (5), CaCl_2_ (1.25), and Mannitol (45), and gassed with 95% O_2_ and 5% CO_2_, with a pH of 7.0, at 37 °C. The hearts were placed in a custom-designed chamber to subdue the motion artifacts, along with a 15 min blebbistatin (5 µM) in perfusate to minimize the contractions. Bolus injections of voltage (RH237, 25 µL of 2 mg/mL dimethyl sulfoxide (DMSO)) and Ca^2+^-indicator dye (Rhod-2/AM, 200 µL of 2 mg/mL DMSO) were injected in the air trap above the aortic cannula. Fluorescence from the epicardium was collected with a camera lens, split with a 570 nm dichroic mirror and focused on two CMOS cameras (Sci-Media UltimaOne, Costa Mesa, CA, USA) capturing the fluorescence emission at 570–595 nm for the cytosolic Ca^2+^ and 610–750 nm wavelengths for the voltage, as described in detail previously [33]. After resting sinus rhythm image acquisition, the hearts were paced at 250 ms with programmed stimulation to measure and compare the conduction velocity and transient duration (90%) of various treatment groups.

### 2.5. RNA Sequencing Studies

Libraries were generated with the Illumina Stranded mRNA Library Prep kit (Illumina: 20040534, San Diego, CA, USA), according to the manufacturer’s instructions. RNA was assessed for quality using a Total RNA 15 nt kit (Agilent: DNF-471-33) on an Advanced Analytical 5300 Fragment Analyzer. The RNA concentration was quantified with a Qubit BR RNA assay kit (Invitrogen: Q10211) on a Qubit 4 (Invitrogen: Q33238). Briefly, 50 ng of input RNA was used for each sample. Following adapter ligation, 15 cycles of indexing PCR were completed, using IDT for Illumina RNA UD Indexes (Illumina: 20040553). Library assessment and quantification were performed using Qubit 1x HS DNA (Invitrogen: Q33231) on a Qubit 4 fluorometer and a HS NGS Fragment kit (Agilent: DNF-474-1000). The libraries were normalized and pooled to 2 nM by calculating the concentration based on the fragment size (base pairs) and the concentration (ng/μL) of the libraries. Sequencing was performed on an Illumina NextSeq 2000, using a P3 200 flow cell (Illumina: 20046812). The pooled library was loaded at 750 pM, and sequencing was carried out with read lengths of 2 × 101 bp, with an average of ~40 million reads per sample. The sequencing data were demultiplexed by Illumina, the on-board DRAGEN FASTQ Generation software v3.8.4 (Illumina, San Diego, CA, USA).

The raw RNA sequencing (RNA-Seq) paired-end reads for the guinea pig (GP) samples were processed through the Mayo RNA-Seq bioinformatics pipeline, RapMap version 1.0.0. Briefly, the raw reads were trimmed with a base quality cutoff of Phred score Q30 using Trimmomatic [34]. RapMap employs the very fast and accurate pseudo aligner and transcript assembler, Kallisto (https://pachterlab.github.io/kallisto/manual, accessed on 10 October 2022), to align the filtered reads to the reference GP genome Cavia_porcellus.Cavpor3.0. Kallisto was also used to calculate the relative expression values across transcripts. Tximport [35] was used to summarize transcript level counts at gene level. Finally, the comprehensive quality assessment of the RNA-Seq samples was performed using MultiQC [36]. The results from all the modules described above are linked through a single html document and reported by the RapMap pipeline.

Using the gene-level counts from RapMap, genes differentially expressed between the control and treated groups were assessed using the bioinformatics package edgeR 2.6.2 [37]. Differentially expressed genes (DEGs) were reported along with their magnitude of change (log2 scale) and their level of significance (False Discovery Rate, FDR < 5%). DEGs were plotted using volcano plots and heatmaps using R Bioconductor packages ggplot2 (https://ggplot2-book.org/, accessed on 10 October 2022) and heat map (https://cran.r-project.org/web/packages/pheatmap/index.html, accessed on 10 October 2022), respectively. Canonical pathway analysis was performed using the Enrichr [38] database. Pathways identified using KEGG were reported. R package pathview (https://pathview.r-forge.r-project.org/, accessed on 10 October 2022) was used to visualize the gene expression changes in pathways of interest.

### 2.6. Statistical Analyses

All continuous variables generated on GP hearts were checked for normality with the skewness and normality test. Variables in each treatment group were compared to controls using an unpaired Student’s *t*-test. A two-tailed *p*-value of <0.05 was considered significant. Stata 16.1 (Stata Corporation, College Station, TX, USA) was used for these analyses.

## 3. Results

### 3.1. CTP Synthesis, Targeting, Conjugation to Amiodarone, and Conjugate Stability

This conjugation was accomplished with a disulfide bond between the quaternary amine salt of amiodarone and the N-terminus of CTP (Figure 1). The rationale behind this approach was to create a conjugate that was stable in serum, but once internalized into cardiomyocytes, the reducing intracellular environment would reduce the bond releasing amiodarone from CTP. This bond was stable for up to 22 days (longest time interval tested) at 37 °C (Figure 2). The rat hearts perfused with Cy5.5 labeled CTP showed robust uptake of CTP (red) by cardiomyocytes and negligible co-localization with WGA-488 or uptake of CTP by myofibroblasts (Figure 3).

### 3.2. Guinea Pig Studies

Adult male GPs (250–350 g) were injected intraperitoneally with amiodarone (80 mg/kg; *n* = 4) daily for 7 days, CTP–amio (26.3 mg/kg; *n* = 4) for 5 days, or CTP alone (17.8 mg/kg; *n* = 4) for 5 days and compared to vehicle-only injected GPs (*n* = 10). At the end of the injection period, the GPs were euthanized, and their hearts were excised and perfused in a Langendorf apparatus. The hearts were labeled with a voltage-sensitive dye (RH 237) and a Ca^2+^ indicator (Rhod-2/AM) with a bolus injection in the perfusate in the air trap above the aortic cannula. Fluorescence images of the heart were focused on 2 CMOS cameras, one to detect voltage and the other intracellular free Ca^2+^. The heart rate, action potentials (APs), and intracellular free-Ca^2+^ transients (CaT) were measured from 10,000 pixels of the voltage and Ca^2+^ CMOS cameras (Appendix A). Durations of the APs and CaTs were calculated from the first derivative of the rise of the signals to their recovery to 90% of the baseline. Amiodarone injections significantly decreased the heart rate during sinus rhythm, decreased the conduction velocity (CV) of the APs and CaT, and increased the AP and CaT durations compared to the control hearts (Figure 4a–e). The conjugate CTP–amio decreased CVs to values similar to amiodarone, and statistically significant decreases compared to the control hearts. However, in contrast to amiodarone, CTP–amio the increased heart rate and decreased the AP and CaT durations, effects that seem to be driven by the CTP component of the conjugate, because the conjugate alone, CTP decreased the AP and CaT duration (Figure 4).

### 3.3. RNA Sequencing Studies

The mRNA was extracted from the 16 GP (4 controls, 4 amio, 4 CTP–amio, and 4 CTP-only treated) hearts, sequenced with 20 million reads per sample, and the reads were aligned to GP cDNA using the Kalisto program for assessing degree of alignment (Appendix A), which showed that ~80% of the reads were successfully aligned (Appendix A). Principal component analysis showed good clustering of the samples within groups and a separation between groups (Figure 5). Because of the unexpected effects of the CTP alone, which was thought to be inert to cardiomyocytes, we also present the comparison of CTP-treated hearts to the control hearts, as well as the amiodarone and CTP–amio treated hearts in the supplementary data (Appendix A). In the CTP-treated hearts compared to controls, there was an upregulation of α-adrenergic receptors and downregulation of β-adrenergic receptors, possibly explaining the increase in the resting heart rate in the CTP- (Figure 6) and CTP–amio-treated hearts (Appendix A). Additionally, the calcium-handling channel genes, *DHPR* and *SERCA2a*, were significantly upregulated (Figure 6), possibly explaining the decrease in late CaTs. Several proinflammatory genes, like *NF-κB*, *TNF-α*, *IL-1β*, and *COX2,* were downregulated in the CTP-treated hearts as compared to the control hearts (Figure 7).

## 4. Discussion

The targeted delivery of therapeutics to the heart remains an elusive goal but one of high clinical impact. In the current body of work, we show the chemical possibility of conjugating amiodarone to a novel, synthetic, and non-naturally occurring cardiomyocyte-targeting peptide or CTP. The conjugation was through a covalent linker containing a disulfide bond, which was stable at 37 °C for up to 22 days. Uptake of this conjugate by GP hearts was indirectly evidenced by the decrease in the voltage and calcium conduction velocities similar to amiodarone alone but at ~1/15th the molar equivalent total dose. CTP–amio had additional effects on the GP hearts, like an increase in the resting heart rates and a decrease in the AP and CaT durations, which were not caused by amiodarone treatment alone. These effects appear to be driven by the CTP portion of the conjugate and were seen with CTP alone injections, to a similar extent. These effects of CTP could potentially be due to changes in the adrenergic receptor and calcium-handling protein expression, as seen in our RNA sequencing experiments. A surprise finding was an anti-inflammatory effect of CTP alone compared to the control hearts. There are multiple literature reports, in cells other than cardiomyocytes, showing that a decrease in cytosolic calcium leads to inhibition of NF-κB activation, which is a transcription factor upstream of TNF-α and IL-1β [39,40,41,42,43]. Based on our data and the existing literature, we hypothesize that CTP increases the expression of calcium-handling genes leading to a decrease in the free cytosolic calcium and a decrease in the NF-κB expression leading to an anti-inflammatory effect.

Cell-penetrating peptides have been studied intensively as vectors for the last 25 years. There is emerging evidence that they, by themselves, may possess antibacterial or antitumor efficacy. However, this observation has been attributed largely to disruption of cell membranes expressing high levels of certain phospholipids or sphingolipids characteristic of tumor cells. We had considered CTP to be an inert cardiomyocyte-targeting peptide. CTP having salutary effects on the calcium handling and suppressing expression of proinflammatory genes was an unexpected finding. Whether this is a dose-dependent effect seen as a result of a fairly high and repetitive dosing of CTP remains to be seen.

CTP has been shown to be cardiomyocyte-specific by independent investigators, although the exact mechanism of transduction remains unknown. In their work, Avula and colleagues showed that placing CTP on a pegylated nanoparticle bearing photosensitizer led to cardiomyocyte-specific ablation with sparing of “innocent bystander” cells like myofibroblasts and endothelial cells, similar to our findings [28]. Additionally, exosomes labeled with CTP and loaded with anti-RAGE (Receptor for Advanced Glycation End-products) siRNA were able to ameliorate myocarditis in a rat model [30]. Although our initial phage display work that led to the identification of CTP was carried out in a rat cardiomyoblast cell line, with subsequent cycles of in vivo phage display in mice [26], CTP is not species-limited to mice, as its efficacy as a vector has been shown in sheep [28] and rat models [30] for delivery of a myriad of cargoes. Indeed, our own lab showed that human heart tissue explanted from patients undergoing heart transplants could be successfully transduced with fluorescently labeled CTP with uptake of the peptide by normal cardiomyocytes while sparing the fibroblasts making up the scar tissue. Cardiomyocyte uptake of CTP was not simply due to an increase in plasma membrane permeability, as demonstrated by a lack of Evans blue uptake [31].

Multiple investigators have tried to target the delivery of amiodarone to the heart. One approach has been to apply amiodarone directly to the epicardium using amiodarone-infused hydrogels [44,45,46] or via atrial patches [47]. This may be of utility under certain circumstances, like patients undergoing coronary artery bypass grafting or valve replacements, but it is not practical for routine long-term treatment. Amiodarone-loaded cyclodextrin nanoparticles efficiently increase cardiac uptake of amiodarone in macrophage abundant tissues such as in the inflamed heart in the setting of myocarditis, resulting in decreased severity of off-target systemic toxicity, but they do not solely target the cardiomyocytes of the non-inflamed heart [48]. In a similar manner, amiodarone-loaded Poly(lactic-co-glycolic acid nanoparticles [49] or liposomal-based amiodarone formulations [50,51] increase the solubility of the lipophilic amiodarone drug resulting in a controlled release that reduces the systemic toxicity but are not exclusive to the targeting of cardiac tissues. Our approach would target delivery specifically to cardiomyocytes for arrhythmia management.

Our work has raised several interesting questions. The study utilized fairly high doses of amiodarone in GPs (80 mg/kg) equivalent almost to tenfold the human loading dose of 400 mg twice daily. Whether CTP will still alter gene expression at lower doses or “maintenance” doses remains to be seen. Our work generated several hypotheses linking CTP to changes in calcium handling and subsequent decrease in NF-κB activation, which are currently under active study. Additionally, the efficacy of CTP–amio in a relevant animal model of afib remains to be seen.

## 5. Conclusions

Targeted drug delivery is the next frontier in modern medicine. In this body of work, we show that amiodarone, a very well established anti-arrhythmic, can be conjugated to a cardiomyocyte-specific cell penetrating peptide for delivery to the heart. Although we saw the expected effects of CTP–amiodarone on channel physiology at 1/15th the dose of amiodarone in an acute administration model, without any overt toxicities (no animal mortality or development of overt toxicities), we saw additional salutary effects on calcium handling and inflammatory markers that appear to be driven by CTP. Our data also suggests that amiodarone use could be expanded by targeted delivery, as it would require significantly lower doses of the drugs, likely leading to less off-target (pulmonary, thyroid, ocular, liver, and skin) toxicities. Our findings are provocative and need further study.

## Figures and Tables

**Figure 1 pharmaceutics-15-02107-f001:**
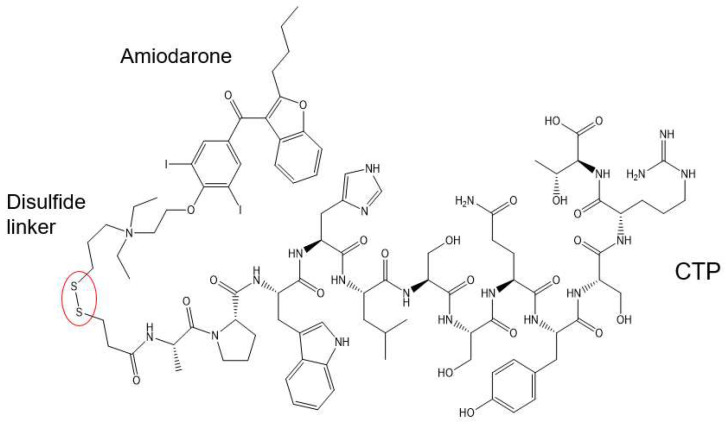
Schematic presentation of the CTP–amiodarone chemical structure. CTP is conjugated to amiodarone via a disulfide linker (circled in red).

**Figure 2 pharmaceutics-15-02107-f002:**
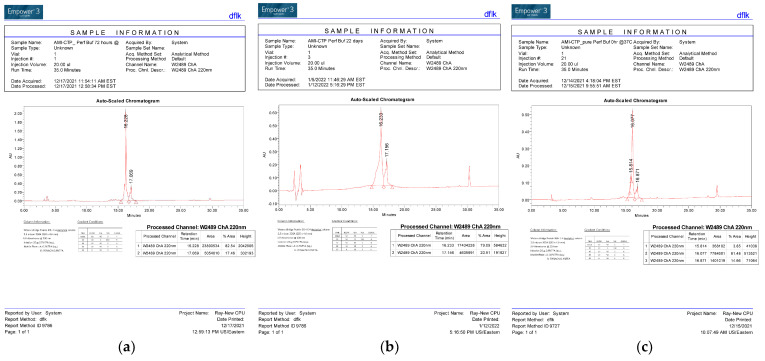
HPLC Tracing of the CTP–amio conjugate immediately upon synthesis and purification (**a**) after 72 h of incubation (**b**) and after 22 days of incubation (**c**) at 37 °C.

**Figure 3 pharmaceutics-15-02107-f003:**
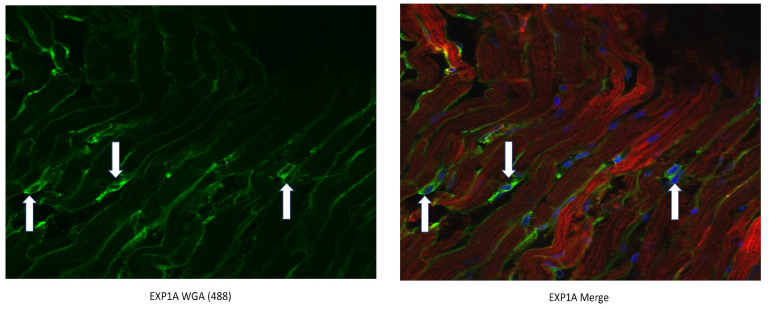
CTP-Cy5.5 transduces rat cardiomyocytes in a cardiomyocyte-specific manner. Myofibroblasts are stained green with WGA-488 (white arrows) and show no co-localization with the CTP-Cy5.5 (red) that is taken up exclusively by cardiomyocytes. Nuclei stained blue (DAPI).

**Figure 4 pharmaceutics-15-02107-f004:**
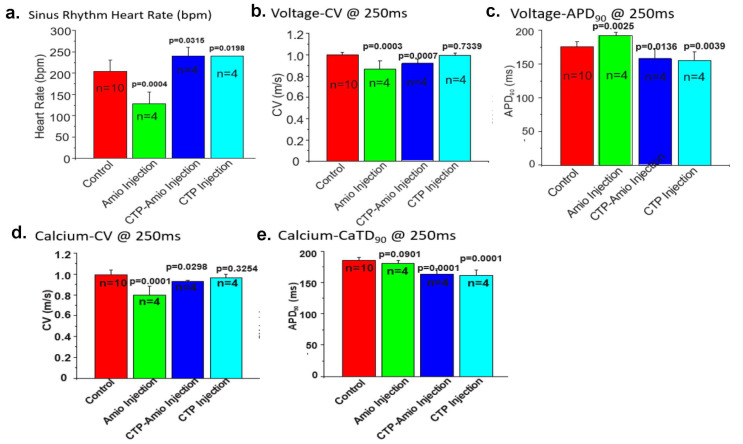
Comparison of the heart rate (**a**), voltage conduction velocities (**b**), action potential duration (**c**), calcium conduction velocities (**d**), and CaT (**e**) between the various treatment groups. All *p*-values are two-tailed unpaired *t*-tests between each treatment group and the controls. (Panel (**b**) is reproduced by permission from Frontiers in Chemistry [32]).

**Figure 5 pharmaceutics-15-02107-f005:**
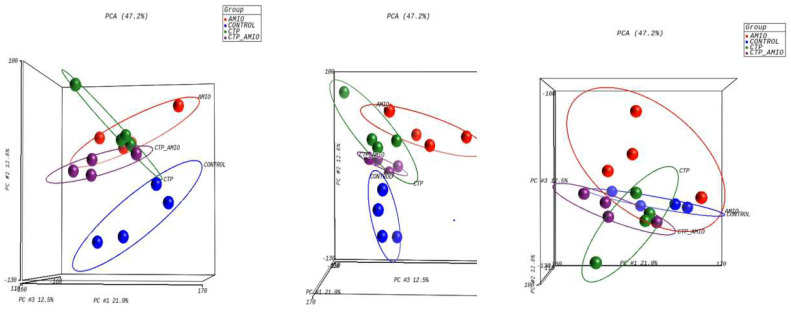
Principal component analysis showing reasonable clustering within groups and separation between different treatment groups.

**Figure 6 pharmaceutics-15-02107-f006:**
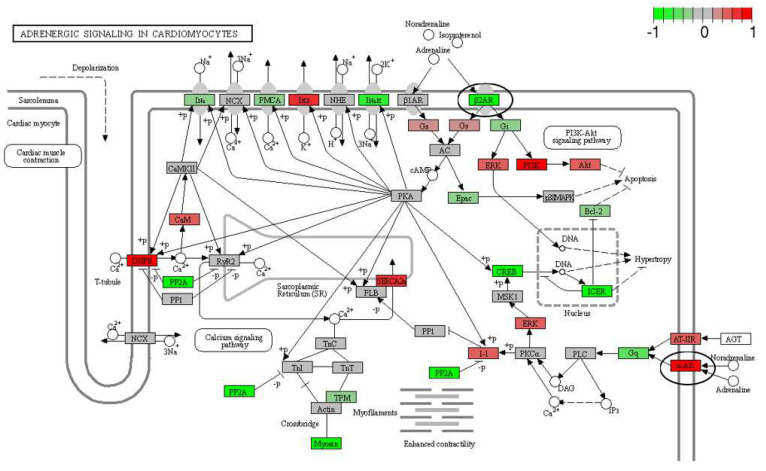
Differentially expressed genes in CTP-treated guinea pig hearts compared with control hearts. The β2-adrenergic receptor is significantly downregulated, whereas the α-adrenergic receptor and the calcium-handling genes (*DHPR* and *SERCA2a*) are significantly upregulated (all circled in black).

**Figure 7 pharmaceutics-15-02107-f007:**
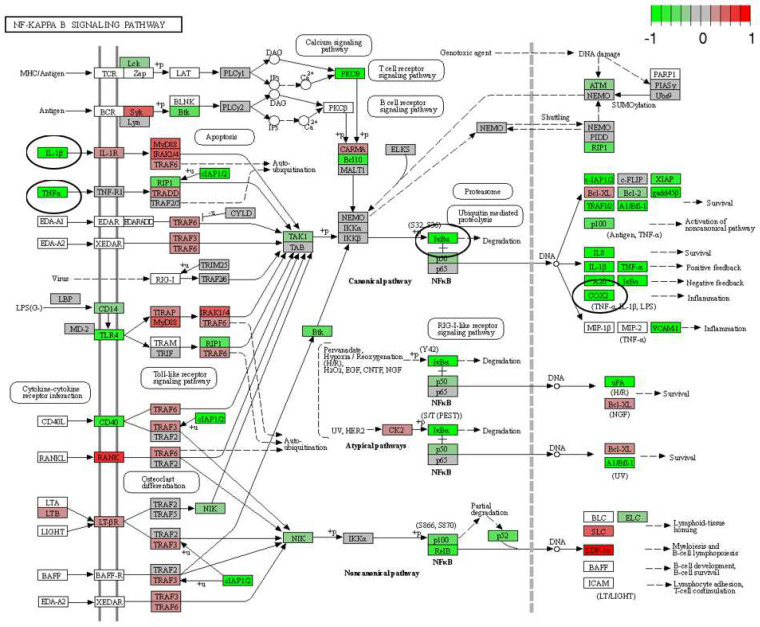
Differentially expressed genes (DEGs) in CTP-treated guinea pig hearts compared with the control hearts. Proinflammatory genes in the NF-κB pathway (*IκBα*), as well as *TNF-α*, *IL-1β*, and *COX2,* are significantly downregulated (all circled in black).

## Data Availability

All data generated form this study is available in the manuscript.

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
