# Peer review of "Cardiomyocyte-Targeting Peptide to Deliver Amiodarone"

_pharmaceutics, 2023, doi:10.3390/pharmaceutics15082107_

Round 1

Reviewer 1 Report

The manuscript written by Zahid et al. is about using a cardiomyocyte targeting peptide conjugated to amiodarone to favor drug delivery to the heart. This topic is a continuation of the research previously published in Front. Chem. 2023 (doi: 10.3389/fchem.2023.1220573). The manuscript is interesting, well-written, and organized. The data is convincing and contains appropriate controls. I think the topic fits within the journal’s scope.

There are some questions/concerns and suggestions that authors should address prior to considering this manuscript for publication.

Minor comments

-          Page 4; Line 124. The formula for diethyl ether is Et2O and not “EtO2”. Please correct it.

-          Page 4; Line 138. The authors state that a MALDI TOF/TOF mass spectrometer was used “to confirm the expected mass and purity of the final compound.” Although the chemistry part was already published, the authors should introduce the corresponding spectra of the conjugate in the supplementary material.

-          Page 6; Line 223. There is a typo “(componenet)”. Please correct it.

 Major comments

-          Figure 2. All HPLC spectra should be moved to the Supplementary information.

-          Figure 4. The authors should use the same letter font and style for both the X- and Y-axis of each of the graphs shown in the figure.

-          In vivo studies should be complemented with a biodistribution study.

-          The authors should include a Conclusion section in this manuscript.

Minor editing of English language required

Author Response

Dear Reviewer,

Thank you for your valuable time and input into our manuscript. The suggested changes have enhanced the quality of our manuscript. We have undertaken the following revisions.

1). All typos/errors/spelling mistakes have been corrected. 

2). The MALDI/TOF reference has been provided, and is an easy, online, free access link. We feel that since it is already published, we do not want to republish the data.

3). The HPLCs in Figure 2 are an important piece of evidence showing the stability of the conjugate over time, which would be needed for long-term applications, and hence is clinically relevant and should be presented up front.

4). We have added a conclusion section.

5). Biodistribution studies on just CTP have already been performed and published, and reference is provided (doi: 10.3390/biom8040147). We agree that a repeat biodistribution study of the CTP-amio conjugate is warranted, we believe that is the next step in our studies, and an entirely new study, for which we are actively seeking funding to carry out. 

Reviewer 2 Report

Here the authors present a study that investigated the effects of a cardiomyocyte targeting peptide (CTP) conjugated to amiodarone, a drug used to treat irregular heartbeats. The authors claimed that CTP-amiodarone could deliver the drug to the heart cells more efficiently and with less side effects than amiodarone alone. They tested their hypothesis on guinea pig hearts and measured various parameters related to cardiac function and gene expression. They found that CTP-amiodarone reduced conduction velocity, action potential duration, and calcium transient duration, while increasing the expression of calcium-handling genes and decreasing the expression of proinflammatory genes.

Major points :

 The text is well-written and concise, but it lacks some important details that would make it more informative and convincing. For example, the authors did not explain how they synthesized and purified CTP-amiodarone, how they injected it into the guinea pigs, and how they controlled for possible confounding factors such as differences in body weight, age etc. The histology is also not convincing. 

They also did not provide proper statistical analysis for their data, which makes it difficult to assess the significance and reliability of their results. The pathway analysis is also lacking, as no overlap or Z scores are calculated to stratify any observed differences. 

Furthermore, they did not discuss the implications of their findings for clinical applications, such as whether CTP-amiodarone could be used to treat human patients with arrhythmias, what dose and frequency would be optimal, and what potential risks or benefits it might have compared to other treatments.

This work would reauire significant revision, to be suitable for publication. 

Fine, require some changes in wording that is sometimes strange. 

Author Response

Dear Reviewer,

Thank you for your insightful comments. We have referenced a very detailed, step-by-step protocol for the synthesis of the CTP-amiodarone conjugate that is available online already which gives the process in thorough detail enabling any scientist to replicate our synthesis if he/she so wishes ( doi: 10.3389/fchem.2023.1220573). Additionally, we have clarified in the methods section that the dose was weight-based (Amiodarone 80mg/Kg intraperitoneal injections daily for 7 days). CTP-amiodarone was at 1/10th the MOLAR dose of Amiodarone (numbers provided in methods section). The comparisons in conduction velocity, action-potential duration was all pair-wise Student's T-test with two-tailed p-values<0.05 considered significant (Statistical Section under Methods). We have also added a statement on clinical implications of our studies under Conclusions.

Reviewer 3 Report

The article is well written and informative, however several modifications should be made before acceptance.

Authors should demonstrate amiodarone release from the conjugate in reducing conditions to show intracellular availability.

Authors need to compare the toxicity of free drug and conjugate at least by IHC on tissues.

The reationale for the NGS studies should be presented in Introduction.

Dosage choices for drug, free peptide and conjugate needs to be justified.

For targeted delivery to cardiomyocytes, Authors should use a fluorescently-labeled drug and compare uptake of free drug and conjugate.

Fig 3 should be completed with statistical analysis.

Author Response

Dear Reviewer,

Thank you for your time and invaluable input. We wanted to clarify a few points:

1). Guinea pigs were chosen instead of mice as they have L-type calcium channels similar to humans allowing for calcium conduction velocities to be measured.

2). The acute dose chosen was for a short term study with dose selected from prior studies in the literature (reference provided).

3). We did not have toxicity studies planned as part of the initial grant application, and hence do not have the funds to carry those out (but planned for future studies/grant applications).

4). CTP by itself has shown no toxicities to date (unpublished data). Amiodarone is an FDA approved Drogin clinical use and has a favorable short-term toxicity profile (significant toxicities long-term and with chronic use). 

5). We have bodistribution studies on CTP alone that have been published. There is no chemical way of fluorescently labeling Amiodarone. We are, however, in the process of developing an LC-MASS spec method of amiodarone/desethylamiodarone extraction from tissues for quantification after prolonged use in future studies.

6). Figure 3 is qualitative, not quantitative, data showing the lack of co-localization of CTP (red) with myofibroblasts (green), confirming the cardiomyocyte-specificity of CTP as others have shown before (doi: 10.1126/scitranslmed.aab3665).

Reviewer 4 Report

1. Letters are different between the manuscript

2. There is no order in the manuscript, all missed up. Please correct.

3. Figures are ok in content, but they need to be improved, missed up.

4. Some references are missing like in this sentence: "Cell penetrating peptides have been studied intensively as vectors for the last 25 years. There is 253

emerging evidence that they by themselves may possess anti-bacterial or anti-tumor efficacy." "CTP has been shown to be cardiomyocyte specific by independent investigators."

5. Discussion is missing concerning how peptides can cross membranes.

6.  What happen to ethical comitee in this cases? Indeed, our own lab showed that human heart tissue explanted from patients 268

undergoing heart transplants could be successfully transduced with fluorescently labeled CTP with uptake 269

of the peptide by normal cardiomyocytes while sparing the fibroblasts making up the scar tissue.

7. There is no ) in this sentence:

Poly(lactic-co-glycolic

8. Experimets are need concernin NF-kB activaction

English need to be improved

Author Response

Dear Reviewer,

Thank you for your invaluable time and input. We have made the following corrections.

1). All fonts are Ariel, 12 for headings, and 11 for the text including the references.

2). All sections have headings (Introduction, Methods, Results, Discussion etc.), and all sections have subsections separating CTP-amio synthesis from the animal studies in both the methods and results sections.

3). Please note that all animal studies were approved by the University of Pittsburgh's animal care and use committee before start of any animal studies, as stated in the "Methods Section/Guinea Pig Studies". There are no human studies that are part of this manuscript, and none were carried out. The human studies referred to in the discussion section were on ex-planted heart tissue from patients undergoing heart transplants, and was deidentified data, with tissue obtained under an IRB protocol. That data forms part of an earlier study that is referenced.

4). The mechanism of transduction of CTP is not known-that is clarified in the discussion section.

5). We agree with the reviewer regarding NF-kappa B. The RNA sequencing data presented is hypothesis generating and the studies to confirm this in vitro in a cardiomyocyte cell line are being planned for future follow-on studies.

Round 2

Reviewer 1 Report

I recommend this manuscript for publication

Reviewer 2 Report

The authors have now answered most points, raised in the first round of review.

Reviewer 4 Report

Thank you for your answer.

Major issues have been resolved.